# Red Blood Cell-Related Phenotype–Genotype Correlations in Chronic and Acute Critical Illnesses (Traumatic Brain Injury Cohort and COVID-19 Cohort)

**DOI:** 10.3390/ijms26031239

**Published:** 2025-01-31

**Authors:** Darya A. Kashatnikova, Alesya S. Gracheva, Ivan V. Redkin, Vladislav E. Zakharchenko, Tatyana N. Krylova, Artem N. Kuzovlev, Lyubov E. Salnikova

**Affiliations:** 1Vavilov Institute of General Genetics, Russian Academy of Sciences, 119991 Moscow, Russia; daria_sv11@mail.ru (D.A.K.); palesa@yandex.ru (A.S.G.); 2Lopukhin Federal Research and Clinical Center of Physical-Chemical Medicine of Federal Medical Biological Agency, 119435 Moscow, Russia; 3Federal Research and Clinical Center of Intensive Care Medicine and Rehabilitology, 107031 Moscow, Russia; iredkin@fnkcrr.ru (I.V.R.); vzakharchenko@fnkcrr.ru (V.E.Z.); tkrylova@fnkcrr.ru (T.N.K.); artem_kuzovlev@mail.ru (A.N.K.); 4National Research Center of Pediatric Hematology, Oncology and Immunology, 117997 Moscow, Russia

**Keywords:** anemia, RBC-related phenotype, traumatic brain injury (TBI), COVID-19, whole-exome sequencing (WES), Notch signaling pathway genes

## Abstract

Changes in red blood cell (RBC)-related parameters and anemia are common in both severe chronic and acute diseases. RBC-related phenotypes have a heritable component. However, it is unclear whether the contribution of genetic variability is pronounced when hematological parameters are affected by physiological stress. In this study, we analyzed RBC-related phenotypes and phenotype–genotype correlations in two exome-sequenced patient cohorts with or at a high risk for a critical illness: chronic TBI patients admitted for rehabilitation and patients with acute COVID-19. In the analysis of exome data, we focused on the cumulative effects of rare high-impact variants (qualifying variants, QVs) in specific gene sets, represented by Notch signaling pathway genes, based on the results of enrichment analysis in anemic TBI patients and three predefined gene sets for phenotypes of interest derived from GO, GWAS, and HPO resources. In both patient cohorts, anemia was associated with the cumulative effects of QVs in the GO (TBI: *p* = 0.0003, OR = 2.47 (1.54–4.88); COVID-19: *p* = 0.0004, OR = 2.12 (1.39–3.25)) and Notch pathway-derived (TBI: *p* = 0.0017, OR = 2.33 (1.35–4.02); COVID-19: *p* = 0.0012, OR =8.00 (1.79–35.74)) gene sets. In the multiple linear regression analysis, genetic variables contributed to RBC indices in patients with TBI. In COVID-19 patients, QVs in Notch pathway genes influenced RBC, HGB, and HCT levels, whereas genes from other sets influenced MCHC levels. Thus, in this exploratory study, exome data analysis yielded similar and different results in the two patient cohorts, supporting the view that genetic factors may contribute to RBC-related phenotypic performance in both severe chronic and acute health conditions.

## 1. Introduction

A critical illness is a condition in which vital organs are damaged, and death is likely without treatment, but recovery is possible. A critical illness describes the condition of a patient beyond the diagnosis of disease [1]. It can develop into an acute or a chronic illness and requires the adequate activation of interrelated physiological responses to provide essential energy, modulate the immune response, and ensure hemodynamic homeostasis to maintain and restore homeostasis in the human body [2]. Interrelated stress responses are limited in the presence of anemia, which is both a consequence and a risk factor for severe disease progression and adverse outcomes in the intensive care unit (ICU) [3,4].

Anemia is a common finding in patients admitted to the ICU. Anemia is defined as a decrease in the concentration of hemoglobin (HGB) and/or the absolute number of red blood cells (RBCs), resulting in an inadequate supply of physiological requirements, and its common diagnostic method is the assessment of HGB levels [5], in which the normal values are ≥130 g/L in men and 120 g/L in women [6]. In the ICU, nearly 60% of patients had serum hemoglobin levels < 120 g/L on admission, 30% had serum hemoglobin levels < 90 g/L, and >95% of critically ill patients were anemic within 3 days of ICU admission [7]. Anemia may be exacerbated or newly developed in critically ill patients during their stay in the ICU due to repeated blood loss (blood sampling, invasive procedures, surgery, etc.), hemodilution, and inflammation, all of which contribute to a decrease in hemoglobin concentration [8]. Severe anemia resulting in decreased oxygen delivery can affect cardiac, renal, metabolic, and cerebral functions in critically ill patients and is associated with worse outcomes during critical illness [3,8,9,10,11]. However, the role of anemia in the prognosis of critically ill patients remains controversial, and even in severe illness, not all patients develop severe anemia [12].

Anemia is typically a multifactorial condition, and several factors, such as the patient’s medical history and underlying pathological processes, should be considered when making the diagnosis [13]. Genetics also contribute to hematological parameters [14]. An individual’s baseline hematological profile is highly heritable [15,16]. Genetic factors account for individual variation in blood cell measurements, and heritability estimates for most hematological phenotypes range from 50 to 90% in twin studies and from 30 to 40% in population studies using an array-based approach [15,17,18]. Most array-derived hematological associations have been obtained in population studies of healthy individuals and are related to common variants, mostly intronic or intergenic, with likely regulatory activity [15,16]. However, rare coding or splice site variants associated with blood cell traits have an average effect size that is nine times greater than that of common variants [19]. The study of rare variants typically requires sequencing rather than array-based technologies [20].

The influence of human genetics on the RBC profiles in severe diseases is not well understood. To the best of our knowledge, there are no published data on the genetics of anemia that develops under critical conditions. In the general population, the influence of genetics is large; however, it is unknown whether this influence is pronounced when RBC-related parameters are affected by physiological stress. Given the association between anemia and adverse outcomes in the ICU, there is a rationale for the awareness of the role of genetics in the development of anemia in critically ill patients. This knowledge, along with other validated risk factors and scales, could be informative for the careful monitoring of patients at a high genetic risk for anemia in the ICU.

To address these questions, we used whole-exome sequencing (WES) data from two cohorts of patients with or at a high risk of a critical illness, namely, chronic patients with moderate-to-severe (ms) TBI sequelae and patients with acute COVID-19, to assess phenotype–genotype correlations with anemia and RBC-related phenotypes at initial, final, and min–max measurements. The choice of cohorts was based on two factors. First, anemia often develops in both conditions, and its association with patient deterioration and adverse outcomes is supported by the literature data in patients with TBI [11] and COVID-19 [3,21]. Second, for our study group, these were convenient, carefully collected, and characterized samples that allowed us to compare the results of the genetic analysis of anemia and RBC-related phenotypes in acute and chronic conditions using a uniform algorithm.

This study was divided into two phases. In the first phase, only phenotypic data, i.e., RBC-related parameters, were considered to assess their biological relevance and concordance with the literature and to look for some common and different features in the cohorts studied. The latter was important for the interpretation of the subsequent results obtained in the phenotype–genotype phase of the study. Given the literature on the influence of RBC-related parameters on outcomes in critically ill patients, we grouped patients in both cohorts according to outcome and additionally stratified TBI patients according to the presence or absence of anemia. In the second phase of the study, we examined RBC-related phenotype–genotype relationships, focusing on the cumulative effects of rare HI variants (acceptor splice, donor splice, stop gained, frameshift, stop-loss, and start-loss) in specific gene sets. It is known that an excess of rare functional variants in large specific gene sets is associated with the severity of the course of a number of psychiatric, neurological, immune, cardiovascular, and cancer diseases [22,23,24,25,26,27]. In our previous studies, we used a similar approach to assess the cumulative effect of rare HI variants in specific gene sets in severe COVID-19 and TBI, with results consistent with those in the literature [28,29,30]. To generate specific gene sets relevant to the study design, we first performed an enrichment analysis in patient strata and found that, in TBI patients with anemia, there was an enrichment in Notch pathway genes, so based on these results, we included Notch pathway genes in all subsequent analyses. Next, we used several resources to select genes based on prior knowledge, specifically Gene Ontology (GO) to select genes functionally associated with phenotypes of interest, Genome-Wide Association Study (GWAS) data to select genes carrying markers associated with phenotypes of interest, and Human Phenotype Ontology (HPO) to select genes associated with inherited, predominantly monogenic diseases of the blood and blood-forming organs. Therefore, in this exploratory study, we tested the hypothesis of the role of rare functional variants in specific gene sets in anemia and RBC-related phenotypes in patients with severe chronic and acute conditions, represented by TBI and COVID-19 patients, respectively.

## 2. Results

### 2.1. Patients

Demographic and clinical data relevant to the phenotypes studied are shown in Figure 1. In both cohorts, patients were classified according to outcome: unfavorable outcome (no change, worsening, death) versus improvement in the TBI cohort, and death versus recovery in the COVID-19 cohort. TBI outcome status was assessed by five neurological scales commonly used in TBI units (Appendix A) and by the consensus opinion of all physicians involved in the patient’s rehabilitation. None of the characteristics differed between TBI patients with different outcomes, including the percentage of patients with moderate/severe anemia. Infectious and/or neurological causes accounted for a high percentage of TBI patients with pulmonary and genitourinary comorbidities. In patients with COVID-19, hypertension and characteristics directly related to outcome, such as severity of illness, ICU stay, higher chest CT severity scores, use of invasive ventilation, and development of sepsis, were more common in patients who died. None of the patients in either cohort were diagnosed with anemia due to a chronic disease.

### 2.2. RBC-Related Parameters in TBI and COVID-19 Patients Stratified by Outcome

Seven RBC-related parameters were analyzed, i.e., RBC count, hemoglobin concentration (HGB), hematocrit (HCT), mean corpuscular volume (MCV), mean corpuscular hemoglobin (MCH), mean corpuscular hemoglobin concentration (MCHC), and red cell distribution width (RDW). Reference information for the parameters studied is provided in Appendix A. The worst values (at any time during hospitalization) were considered minimum values for all parameters except RDW, for which maximum values were considered unfavorable; we used the term min–max in this context. First, the final and min-max values were compared between different outcome groups in TBI patients (Figure 2A) and COVID-19 patients (Figure 2B). In TBI patients, there were no significant differences in any of the first and final RBC-related parameters. Regarding min–max parameters, MCHC and RDW values were marginally different, with no significant results after correction for multiplicity. Of the 50 patients, 41 had HGB levels below the sex-adjusted reference values at the first and final assessments, and 21 and 17 had HGB levels <100 g/L at the first and final assessments, respectively. In COVID-19 patients, MCHC and RDW values showed significant differences between the analyzed groups at the first, final, and min–max measurements, with the most significant differences for min–max values. Among other parameters, RBC, HGB, and MCV differed at final measurements, and RBC and HGB also differed when min–max values were analyzed. Two patients were diagnosed with moderate anemia on admission, but ten patients had a HGB level of <100 g/L at the final assessment.

### 2.3. RBC-Related Parameters and Neurological Scale Scores in TBI Patients with and Without Anemia

The characteristics of patients with TBI according to the presence or absence of anemia are shown in Appendix A. Of the 50 patients with TBI, 25 had anemia, most commonly iron deficiency anemia (*n* = 19, 76%). Two patients with anemia received red blood cell transfusion. Patients with and without anemia did not differ with respect to sex, age, TBI-related conditions, or comorbidities.

TBI patients with anemia had more unfavorable scores on four of the five neurological scales (except GCS) at baseline compared to TBI patients without anemia. For the FOUR scale, these effects were maintained at the final assessment (Figure 3A). RBC, HGB, and HCT were lower in TBI patients with anemia, but only minimal values of these parameters retained significance after correction for multiplicity (Figure 3B). Correlation analysis of RBC parameters and neurological scale scores showed a weak correlation between lower RBC, HGB, and HCT levels and unfavorable neurological scores at the first assessment and a moderate and significant correlation between them at the final assessment (Figure 3C).

### 2.4. WES Data Overview

In the TBI sample, of the total number of unique variants, i.e., variants with unique identifiers (position and allele), there were 144,249 variants, of which 2911 (2.02%) were HI variants, 52,637 (36.49%) were missense variants, and 43,813 (30.37%) were synonymous variants. In the COVID-19 sample, out of a total of 105,294 unique variants, 3328 (3.16%) were HI variants, 53,348 (50.67%) were missense variants, and 41,560 (39.47%) were synonymous variants. The distribution of variants by allele frequency (AF) was similar, and the number of missense and synonymous variants decreased with decreasing AF. Among the HI variants, those with unknown AF (absent in GnomAD) represented the largest sets (Figure 4A). Among the singletons, common variants (AF > 0.01) accounted for a small proportion, especially among HI variants, which were strongly dominated by variants without AF data (Figure 4B). The distribution of SNVs, insertions, and deletions by AF was also similar; the proportion of variants without AF data was comparable in both sets of variants (Figure 4C). We further combined variants with AF < 0.001 and no AF data under the name “rare” variants. Rare HI variants are called qualifying variants (QVs). The TBI and COVID-19 sets matched 15 QVs (Figure 4D) and 156 genes, respectively, with these variants (Figure 4E).

In the exome-wide analysis of the association of individual variants with anemia in the TBI sample (25 of 50 patients) (Figure 4F) and with a final HGB level of <100 g/L in the COVID-19 sample (ten of 77 patients) (Figure 4G), there were no significant associations of individual variants. The quantile–quantile (QQ) plots showed a deflation of the observed *p*-values, indicating that our sample size was insufficient to allow for analysis and proper interpretation of the results at the level of individual genetic variants (Figure 4F,G).

### 2.5. Gene Set Analysis

In the first step of the gene set analysis, we performed a functional enrichment analysis using STRING version 12.0 with the “Proteins with Values/Ranks–Functional Enrichment Analysis” module for a set of genes with QVs in the four TBI strata: (1) unfavorable outcome, (2) improvement, (3) with anemia, and (4) without anemia (Figure 4A). Values/ranks were assigned based on the number of QVs per gene in the set under consideration. Only in TBI patients with anemia did the enrichment analysis yield a single significant result for the KEGG hsa04330 Notch signaling pathway with four genes, including *LFNG* (14 QVs in ten patients), *ATXN1* (seven QVs in seven patients), *DTX3L* (one QV in one patient), and *NCOR2* (ten QVs in 12 patients). For comparison, QVs were also observed in three of these genes in non-anemic patients with TBI: *LFNG* (eight QVs in five patients), *ATXN1* (two QVs in two patients), and *NCOR2* (seven QVs in seven patients). The high abundance of these genes in TBI patients is probably explained by the nature of the sample, for which we have previously demonstrated a strong enrichment of developmental and/or nervous system-related genes, including *LFNG*, *ATXN1*, and *NCOR2* [30]. Functional enrichment analysis of the COVID-19 outcome strata (1) death and (2) recovery did not yield significant results (Figure 4A).

In the second step, we generated four sets of genes with QVs in two samples. Three sets of genes were constructed based on the existing knowledge of genes involved in the development and function of the hematopoietic system: (1) associated with GO terms for the keywords hemopoiesis, erythrocyte, leukocyte, and platelet, and this set was called the GO-derived gene set; (2) associated with blood count phenotypes (both common and rare variants) from the largest genetic studies in individuals of European ancestry [31,32], and this set included red blood cell (RBC)-, white blood cell (WBC)-, and platelet (PLT)-related genes and was termed the GWAS-derived gene set; and (3) associated with inherited abnormalities of blood and blood-forming tissues, abnormal erythrocyte morphology and abnormal erythrocyte physiology, abnormal leukocyte morphology, abnormal leukocyte count, and abnormality of thrombocytes from the Human Phenotype Ontology (HPO) resource, and this set was called the HPO-derived gene set. Since the genes associated with different blood phenotypes within the gene set overlapped strongly (Figure 4B), our subsequent analysis included genes from the full sets without distinction of blood cell types. The fourth gene set was constructed based on the results of the functional enrichment analysis and included genes with QVs from the KEGG hsa04330 Notch pathway (*n* = 52). The number of genes and their overlap in the generated sets in the TBI and COVID-19 samples are shown in Figure 4C and Appendix A, respectively.

In the third step, we performed an association analysis of the cumulative effects of the QVs within the derived gene sets for anemia in the TBI sample and for reduced HGB levels below the threshold < reference level (Appendix A) and <100 g/L in the COVID-19 sample (Figure 5D). In the TBI sample, the GO-derived and hsa04330 gene sets appeared to be associated with anemia. In the COVID-19 sample, a marginally significant effect was observed for the hsa04330 gene set when the threshold for HGB was <the reference level, but below the threshold for HGB < 100 g/L, pronounced effects were observed for the same gene sets (GO-derived and hsa04330) as in the TBI sample. The wide 95% CI reflects the sample statistics (hsa04330 gene set, seven QVs in six genes) and is consistent with the literature data in a similar setting [33].

### 2.6. Effect of Genetic Variable on the RBC-Related Parameters

To assess the contribution of the QVs in the studied gene sets to the RBC-related parameters, we performed multiple linear regression (MLR) analysis by adding the genetic component to a number of independent variables, i.e., sex, age, and clinical status according to the outcome. The percentage of variance explained (R-squared) in the TBI sample (Figure 6A) reached significant values for RBC and HGB and partially significant values for HCT at the final and min–max measurements, with no influence of genetic variables on these dependent variables. As for the other parameters, the results were inconsistent with the most significant effects registered for min–max values. Significant *p*-values for independent genetic variables, i.e., the number of QVs per person in gene sets, were found for the final and minimum measurements of MCV and GWAS-derived gene set, MCH and GO-derived gene set, minimum MCH and hsa04330 gene set, and minimum MCHC and GO-derived gene set (Figure 6B). The increase in the number of QVs correlated with the decrease in these parameters, thus contributing to their unfavorable levels.

The percentage of variance explained in the COVID-19 sample (Figure 6C) reached significant values for most RBC-related parameters and at different time points, except for MCV and MCH. The highest R-squared value was observed for MCHC at the final and minimum measurements. In contrast to the TBI sample, the hsa04330-derived gene set was a significant contributor to RBC, HGB, and HCT in the COVID-19 sample. Other genetic variables influenced MCHC values at different time points; however, most effects were registered for the minimum values (Figure 6D).

## 3. Discussion

In this study, we analyzed RBC-related phenotypes and phenotype–genotype correlations in two patient cohorts: patients with msTBI hospitalized for rehabilitation and patients with COVID-19. In the TBI cohort, most of the differences in RBC-related measures between the patient groups defined by outcome or presence/absence of anemia were small. The exceptions were highly significant differences in minimum RBC, HGB, and HCT levels in patients grouped by anemia diagnosis. TBI patients with anemia had worse neurological scale scores, which correlated with RBC, HGB, and HCT levels. COVID-19 patients grouped by outcome showed contrasting values for many RBC-related parameters. Highly significant differences between deceased and recovered patients were found for MCHC and RDW levels at all assessments, whereas most other parameters, including HGB levels, differed at the final and min–max assessments. Analysis of the WES data, focusing on rare HI variants (QVs) in specific gene sets, showed that genetics contributed to both differences and similarities in RBC-related phenotypic characteristics in the study cohorts. Cumulative effects of QVs were observed in gene sets associated with GO terms related to the blood phenotype and Notch signaling pathway in TBI patients with anemia and in COVID-19 patients with HGB levels <100 g/L at the final measurement. In the MLR analysis, the number of rare HI variants per person in gene sets associated with RBC-related phenotypes contributed to the final and/or minimum MCV, MCH, and MCHC levels in patients with TBI. In COVID-19 patients, QVs in Notch signaling pathway genes influenced minimum RBC, HGB, and HCT levels, and genes from other sets influenced MCHC levels at different time points.

Traumatic brain injury (TBI) has acute and long-term pathophysiological consequences, including anemia. Anemia in patients with TBI in the acute period is attributed to blood loss during trauma and surgery and is associated with increased apoptosis and impaired oxygen extraction, increased cerebral blood flow and intracranial pressure, and impaired cerebrovascular response to changes in CO_2_ levels, leading to cerebral ischemia and worsening brain damage [34]. In the long term, other factors influence the development and severity of anemia. TBI causes chronic activation and progressive dysfunction of the bone marrow stem/progenitor cell pool, leading to irreversible deficits in hematopoiesis, innate immunity, neurological function, and the premature senescence of multiple cell types. Patients with TBI are susceptible to bacterial and viral infections that exacerbate cellular senescence, stem cell depletion, and altered intercellular communication [35,36,37]. In our sample of TBI patients hospitalized for rehabilitation, the majority had decreased hemoglobin levels. Among the patients grouped by the diagnosis of moderate/severe anemia, differences in RBC, HGB, and HCT levels were associated with differences in neurological scale scores. We did not observe marked differences in RBC-related parameters in TBI patients in relation to such a complex and multifactorial feature as an outcome; however, as a result of anemia therapy and rehabilitation, better final RBC, HGB, and HCT levels were associated with better final neurological scores, which is consistent with the literature data [38].

In COVID-19 patients, the most pronounced associations with fatal COVID-19 were found for decreased HGB and MCHC levels, as well as increased RDW levels, which correlate with the literature data [39,40,41,42,43]. In COVID-19, massive exposure to inflammatory mediators results in erythrocyte destruction and decreased erythropoiesis [44,45], which can lead to anemia. Furthermore, due to the role of iron in viral replication, decreased iron bioavailability is another mechanism for the development of anemia in viral infections [46]. Extremely limited oxygen delivery caused by dysfunctional red blood cells may contribute to the development of hypoxia-induced multi-organ damage, leading to fatal outcomes in critically ill patients with COVID-19 [47]. High RDW was also strongly associated with death in the COVID-19 cohort; however, RDW is a non-specific marker of overall disease and all-cause mortality in critically ill patients [48,49]. The association between elevated RDW and COVID-19 mortality may indicate that RBC production kinetics are slowed under conditions of increased WBC and platelet kinetics due to inflammation [48].

In a broader physiological context, critical states such as sepsis, trauma, and shock are characterized by decreased RBC mass and structural and functional changes in RBCs. Inflammatory mediators play an important role in erythropoiesis deficiency and impaired iron metabolism. In addition, the proinflammatory milieu contributes to both functional and structural changes in RBCs, reducing their deformability, and possibly impairing microvascular perfusion. A critical illness is associated with a decreased response to erythropoietin, which binds to erythroid progenitor cells and promotes their proliferation, as well as alterations in iron metabolism and dysregulation of bone marrow function. In severe hypoxia, erythropoietin production can increase 1000-fold, but in critical illness, the erythropoietin response to low hemoglobin levels is impaired [50], especially in patients with sepsis and multiple trauma [51,52]. Cerebral tissue represents only 2% of the body weight, but the high metabolic activity of neural tissue requires 20% of the body’s oxygen supply [53], and it is not surprising that its deficiency is closely associated with deterioration in patients with TBI.

The most interesting finding of the present study was the association of Notch signaling pathway genes with anemia in both cohorts. In the MLR analysis, Notch pathway genes influenced minimal RBC, HGB, and RBC levels in COVID-19 patients, but not in TBI patients, which may be explained by the limited influence of the genetic component in the context of a depleted hematopoietic system resulting from a prolonged severe illness in TBI patients. In addition, since most TBI patients were admitted with decreased HGB levels, corrective therapy was applied, which affected the minimum and final levels of HGB and its closely related parameters.

Notch signaling plays a critical role in regulating the proliferation and differentiation of neural and hematopoietic stem and progenitor cells during embryogenesis and organogenesis [54]. The role of Notch in adult hematopoiesis has long been debated. One study [55] concluded that canonical Notch signaling is dispensable for adult steady-state and stress myeloerythropoiesis. However, subsequent studies have provided evidence supporting the critical role of canonical Notch signaling in stress-induced hematopoiesis [56,57]. In the adult hematopoietic system, Notch receptors and targets are expressed at different levels in different hematopoietic cell types and are thought to influence lineage commitment, including erythroid differentiation [58,59,60]. Disruption of Notch signaling results in reduced pool size, loss of quiescence, altered niche partitioning, and increased recruitment of hematopoietic stem/progenitor cells [61]. Notwithstanding the controversial role of Notch signaling in adult erythropoiesis, it can be argued that the embryonic origin of the adult hematopoietic system determines its capabilities and limitations under critical conditions, with genes of the Notch signaling pathway contributing to these capabilities and limitations. Since a critical illness is a state of severe physical stress in the human body, we hypothesized that genetically mediated disruption of canonical Notch signaling in stress-induced hematopoiesis may have a causal relationship with anemia in both cohorts.

A closer look at the Notch pathway genes revealed genetically mediated interweaving of the neural and hematopoietic systems. In a cohort of TBI patients, four genes were involved in the Notch pathway. Among them, *ATXN1* and *NCOR2* were associated with GO terms related to nervous system development and/or function and blood-related phenotypes in the GWAS. *LFNG* was associated with the GO term hematopoiesis. In an experimental mouse model, Lfng-mediated Notch signaling was shown to be a key factor in regulating neural stem cell quiescence, division, and fate [62]. In the COVID-19 cohort, six genes were part of the Notch signaling pathway. Among them, four genes, *DLL3*, *HDAC2*, *KAT2A*, and *NUMBL*, were associated with nervous system-related GO terms, and another gene, *MAML3*, affected the generation of pyramidal neurons [63]. *KAT2A* was also involved in hematopoiesis and the GWAS of blood phenotypes. These results show that the involvement of anemia-associated genes in nervous system processes in our study may be of interest in the context of the hypothesis that there are molecular networks involved in both the brain and hematopoietic stem cells (HSCs) [64]. In addition to direct niche innervation, HSCs can be under neural control in a variety of ways, including central nervous system and hormone release, neural crest-derived mesenchymal stem cells, and hematopoietic cells that express neuronal receptors and neurotransmitters [65].

Another set of genes that showed an association with anemia and hemoglobin levels <100 g/L in TBI and COVID-19 patients, respectively, was a set of genes associated with biological processes related to hematopoiesis and red blood cell phenotypes according to GO. We used GO, GWAS, and HPO to generate gene sets. The genes from the listed sets partially overlapped and differed in the degree of involvement and significance for the phenotypes in question, but in general, it can be assumed that the GO-based gene selection yielded gene lists enriched for causative genes in both patient cohorts. None of the patients in our study had inherited blood disorders, which is consistent with the marginal influence of HPO-derived genes on RBC-related phenotypes (reported for one index (MCHC) in patients with COVID-19).

Another finding of this study was the role of genetic variables in RBC indices. Genetic variables influenced the final and/or minimum, but not the first, measurements of MCV, MCH, and MCHC in patients with TBI. This is likely due to the fact that the phenotypic performance of the RBC-related indices at baseline included many confounders that were excluded or corrected during treatment, making the follow-up indices more susceptible to genetic factors. In COVID-19 patients, genetic variables strongly influenced the MCHC levels. To explain this effect, we propose the following hypothesis: under physiological stress, blood rheology changes with decreased red cell deformability and increased aggregation, which affects red cell indices, especially MCHC [66]. Because rheological parameters (e.g., RBC aggregation index, RBC stiffness, and deformability) are mediated by different genetic loci [67], the spectrum of genes associated with different RBC indices may differ. MCHC is calculated on the basis of three RBC parameters (HGB, RBC, and MCV) [68] and is affected by changes in any of these parameters. At the same time, multiple processes affecting MCHC can counterbalance each other, making this index vary within a narrow range and relatively stable [69,70], ensuring its sensitivity to genetic variables. The large variation in other measures may explain the lack of significant findings [66].

Rare variant research in common diseases has two main applications: (i) understanding the disease or trait at the molecular and cellular level and (ii) generating informed predictions that may be clinically relevant [71]. In addition, genetic analysis may lead to better phenotypic definitions. For example, some clinical diagnoses may involve a combination of different biological phenotypes with different genetic and cellular pathways that have similar clinical manifestations. In such cases, it may be preferable to study each sub-phenotype separately [72]. Conversely, some diagnoses that are considered distinct may share a common biological origin, making them more appropriately defined as a single phenotype. Our data on Notch signaling genes link unsuccessful neurological rehabilitation and anemia in TBI patients, predicting both conditions within a single phenotype and contributing to a unified approach for developing new therapeutic options.

Clinically relevant predictions based on the results of this study may include decisions regarding the use of soluble iron in combination with erythropoietin and vitamin B12 or transfusion of packed red blood cells (pRBCs) in patients with anemia. Both procedures have their advantages but are also associated with various contraindications and should therefore be used with caution [73,74]. WES or targeted sequencing results for rare/new HI variants in Notch signaling and hematopoiesis-related genes may be useful in determining the appropriateness of pRBC transfusion, as the genetically mediated features of hematopoiesis may be associated with a limited effect of erythropoietin-based therapy. The situation is even clearer when sequencing reveals variants in some specific genes, such as the erythropoietin receptor, *EPOR*, which is a GO hematopoiesis-related gene. Due to HI variants in *EPOR*, erythropoietin may not initiate its internalization. Therefore, blood transfusion should be the better choice.

This study has limitations. The study is limited by sample size and cohort heterogeneity and may have been subject to type I error, i.e., false-positive results in rare variant association studies. The sample size may contribute to bias in the interpretation of certain results for both a specific set of genes and a specific phenotype, and does not allow for the study of individual variants or individual genes. The lack of replication data and independent validation cohorts weakens the reliability of our conclusions and renders them speculative. Given these limitations, we consider our study to be a pilot study. The results are preliminary and represent hypotheses that should be confirmed in other studies with larger cohorts.

## 4. Materials and Methods

### 4.1. Participants

#### 4.1.1. TBI Cohort

Participants in the TBI cohort were recruited between June 2023 and May 2024 from among patients undergoing rehabilitation after msTBI at the Federal Research and Clinical Center of Intensive Care Medicine and Rehabilitation (Moscow, Russia). The study included 50 unrelated patients of European ancestry with a history of msTBI (39 males and 11 females) who were initially treated in different clinics and hospitalized for rehabilitation at different time points (9–526 days) after the accident. Exclusion criteria were age <17 years, mental and neurological diseases that may affect rehabilitation outcomes (e.g., stroke, tumor, etc.), developmental disorders, hereditary neurological diseases, pregnancy, terminal incurable chronic diseases, alcohol- or drug-related injuries, and war injuries. The Institutional Review Board (IRB) of the Federal Research and Clinical Center of Intensive Care Medicine and Rehabilitation approved the study protocol (#2/23/4, 30 May 2023). All the enrolled patients or their legal representatives signed a written informed consent form.

Patients were divided into two groups according to the outcome: improved (*n* = 25) and unchanged, worsened, or died (*n* = 25). Outcome measures included the comparison of patient status at the first and final assessments before discharge, transfer, or death using common brain injury assessment tools, such as the Glasgow Coma Scale (GCS), Coma Recovery Scale-Revised (CRS-R), Disability Rating Scale (DRS), Full Outline of UnResponsiveness (FOUR) scale, and modified Rankin Scale (mRS) (Appendix A) [75,76,77,78,79]. The consensus opinion of the rehabilitation physicians regarding the improvement in the patient’s condition due to rehabilitation was based, among other things, on the results of the neurological scales, i.e., moderate improvement on 3–4 scales or marked improvement on 1–2 scales without deterioration on other scales.

Of a total of 50 TBI patients, 25 had moderate (HGB: 80–109 g/L) or severe (HGB: <80 g/L) anemia [6]. Among the anemic patients, there were no patients with anemia caused by bleeding or anemia caused by specific chronic (e.g., autoimmune diseases, cancer, chronic kidney disease, or long-term infections) or hereditary diseases.

#### 4.1.2. COVID-19 Cohort

A total of 77 patients with COVID-19 from our cohort of 86 patients (45 males and 33 females) with WES data [28] were selected for the study. Patients were recruited in 2020 during the prevaccination period. Exclusion criteria were incurable terminal disease, immunodeficiency (primary or acquired), prolonged use of corticosteroids, pregnancy, alcohol and drug abuse, and HIV/AIDS. From the 86 patients, we also excluded non-Europeans identified by PCA [28] and patients without outcome data. At enrollment, all patients were tested by PCR for SARS-CoV-2 using nasopharyngeal swabs. Diagnosis and severity of illness were determined according to the International Recommendations for the Prevention, Diagnosis and Treatment of Emerging Coronavirus Infection [80] and the World Health Organization’s “Interim Clinical Recommendations for the Prevention, Diagnosis and Treatment of COVID-19-2020”, Ministry of Health of the Russian Federation (BMP_COVID-19_V17.pdf (minzdrav.gov.ru)). At enrollment, two patients were diagnosed with moderate anemia (HGB: 80–109 g/L), but ten patients had an HGB level of <100 g/L at the final assessment. The Institutional Review Board (IRB) of the Federal Research and Clinical Center of Intensive Care Medicine and Rehabilitation approved the study protocol (AE 2.1.18). All the enrolled patients or their legal representatives signed a written informed consent form.

### 4.2. WES and Variant Calling

The WES and variant calling for the TBI cohort were described in [30]. Briefly, blood was isolated using a QIAamp DNA Blood Mini Kit. Sequencing was performed at LLC “Eugene” (Moscow, Russia). DNA was fragmented to an average length of 250 bp using an S220 Focused Ultrasound Fragmentation Device (Covaris, Woburn, MA, USA). Enrichment was performed according to the RSMU_exome protocol [81] by hybridization with the Agilent SureSelect Human All Exon v8 probe set (target size: at least 35 million bp). Enriched DNA library pools were circularized and sequenced in paired-end mode on the MGISEQ-2000 platform using the DNBSEQG400RS High-Throughput Sequencing Kit PE100 according to the manufacturer’s protocol (MGI Tech, Mentor, OH, USA). FastQ files were generated using the zebracall-V2 software (MGI Tech). Bioinformatic processing of the sequencing data for each sample included alignment to the GRCh37 reference genome using bwa mem2 v2.2.1 and SAMtools v1.9, obtaining quality metrics for exome enrichment using Picard v2.22.4, variant calling using bcftools mpileup v1.9 and Strelka2 v2.9.2 software, variant annotation using the web version of the ANNOVAR software (https://annovar.openbioinformatics.org/en/latest/, accessed on 17 June 2024), and a number of custom scripts to improve the quality of the resulting variant annotation files [81]. The MultiQC v1.14 software was run as a final quality control step upon the completion of the bioinformatics pipeline. Variant calls required at least 10× coverage. The sequencing depth was 116.96 ± 55.05 (mean ± SD).

The WES and variant calling for the COVID-19 cohort were described in [28]. Briefly, DNA was isolated from the blood using the Qiagen DNA Blood Mini Kit. Forty-one samples were sequenced at Genomed (Moscow, Russia). The Swift 2S^®^ Turbo DNA Library Kit was used for DNA fragmentation and barcoding. Enrichment was performed using the Twist HumanCoreExome (https://www.twistbioscience.com/products/ngs/fixed-panels/human-core-exome (accessed on 22 June 2022)). Sequencing was performed on an Illumina HiSeq X Ten platform with 150 bp paired-end reads. Thirty-six samples were sequenced at the Resource Center “Bio-bank Center”, Research Park, St. Petersburg State University (St. Petersburg, Russia). Samples were prepared for sequencing using the Illumina TruSeq DNA Exome Kit (https://www.illumina.com/products/by-type/sequencing-kits/library-prep-kits/truseq-exome.html (accessed on 22 June 2022)) and sequenced on HiSeq2500/HiSeq4000 platforms with 90 bp paired-end reads. Reads were aligned to the GRCh38 reference genome using BWA MEM [82]. Duplicate reads were flagged and excluded using the MarkDuplicate program. The target regions in the analysis were the genomic regions common to both sets used for sample preparation. The quality of the mapping results and targeted enrichment was evaluated using the CollectHsMetrics program. Variant calling was performed using the HaplotypeCaller program in the GATK package. Variants were required to meet GATK’s standard variant quality score recalibration (VQSR) threshold with additional filters [28] and to have at least 10× coverage. The sequencing depth was 76.74 ± 56.63 (mean ± SD).

### 4.3. Annotation of Variants

Variant annotation was performed using the AnnoVar and Ensembl Variant Effect Predictor (VEP) (release 112) tools and the Genome Aggregation Database (GnomAD) v2.1.1 population database. Variants judged by Ensembl to have severe consequences for protein structure and function (acceptor splice, donor splice, stop gained, frameshift, stop-loss, and start-loss) were classified as high-impact (HI) (https://www.ensembl.org/info/genome/variation/prediction/predicted_data.html (accessed on 22 September 2024)). Our analysis focused on rare variants with alternative allele frequencies (AF) <0.001 or no AF data (missing) in the GnomAD database and with ≤3 alleles in the TBI sample and ≤4 alleles in the COVID-19 sample to match the observed minor AF ≤ 3% [83].

### 4.4. Compilation and Analysis of Gene Sets

We generated lists of genes that may be important in the context of red blood cell-related phenotypes. A first set of genes was generated by functional enrichment analysis using STRING version 12.0 with the “Proteins with Values/Ranks–Functional Enrichment Analysis” module for a set of genes with rare HI variants present in TBI patients with anemia, where values/ranks were set for the number of unique variants per gene. Since this analysis revealed enrichment of the KEGG pathway hsa04330 (Notch signaling pathway), to further analyze all patient strata for categorical and quantitative data, we extracted the full set of genes within hsa04330 using the Pathway/Process/Disease module of the STRING database version 12.0.

Three other gene sets were generated based on the existing knowledge of genes encoding proteins that may be closely related to hematological parameters. (1) The AmiGO 2 tool (https://amigo.geneontology.org/amigo/landing, accessed on 18 June 2024) was used to retrieve genes associated with GO terms related to hemopoiesis, erythrocyte, leukocyte, and platelet within the “Biological Process” domain in taxon 9606 Homo sapiens. (2) GWAS data for genetic variants associated with blood count phenotypes from the UK Biobank and Blood Cell Consortium (*n* = 563,085 participants of European ancestry) [31] and data from a whole-exome imputation study for very rare coding variants associated with blood cell indices (*n* = 459,259 participants of European ancestry) were used to select genes associated with hematological profiles [32]. These genes comprised a GWAS-derived gene set and included red blood cell (RBC)-, white blood cell (WBC)-, and platelet (PLT)-related genes. (3) The Human Phenotype Ontology (HPO) tool (https://hpo.jax.org/, accessed on 25 August 2024), which provides a standardized vocabulary for phenotypic abnormalities found in human disease, was used to map genes for inherited abnormalities associated with abnormalities in blood and blood-forming tissues, abnormal red cell morphology, abnormal red cell physiology, abnormal leukocyte morphology, abnormal leukocyte count, and thrombocyte abnormalities.

The AmiGO 2 tool was also used to search for genes associated with the terms nervous system development and nervous system process to further discuss genes of the Notch pathway.

### 4.5. Data Analysis

Calculations were performed using the R software (version 3.4.1). Categorical variables were tested using Fisher’s chi-squared test with Yates’ continuity correction to determine whether the proportion of the variable of interest was the same across the compared samples. The likelihood ratio of chi-squared test was used when the number of test levels was >2 for categorical variables (chest CT severity scores and sepsis types). For continuous variables, the normality of the distribution was first tested using the Shapiro–Wilk test. The normality test showed violations of normality for some of the data samples, so we next used the nonparametric Mann–Whitney U test (MWU test) to test whether the two samples being compared were pooled from the same sample. Spearman’s correlation coefficients were used to describe the strength and direction of the relationship between the variables being compared.

Association analysis of rare variant series was performed using the unadjusted two-tailed Cochran–Mantel–Haenszel (CMH) test, which provides a combined *p*-value and odds ratio [25]. The CMH statistic extends Fisher’s exact criterion beyond 2 × 2 stratified contingency tables to test whether the common odds ratios across strata are equal to 1 [84]. This test is preferred to logistic regression because of its robustness and avoidance of inflated test statistics in situations with small numbers of genetic variant carriers [85].

MLR was used to analyze the effect of the independent variables on the effect of the dependent variable, i.e., RBC-related parameters. The independent non-genetic variables included age, sex, and clinical status. In the TBI sample, the clinical status at the first measurement was defined as the GCS score at the first assessment, and at the final and minimum measurements as an unfavorable outcome or improvement. In the COVID-19 sample, clinical status at the first measurement was defined as COVID-19 severity at admission and death or recovery at the final and minimum measurements. The independent genetic variable was defined as the cumulative number of rare HI variants per person in the selected gene set, under the dominant inheritance model. Sex and clinical condition are categorical variables; therefore, they were included in the MLR as dummy variables. The independence of the residuals was assessed using the Durbin–Watson test and was within an acceptable range of 1.50–2.50. The assumptions of homoscedasticity (Breusch–Pagan test *p*-value > 0.05) and multicollinearity (Variance Inflation Factor (VIF) < 5) for the regression analyses were met. The normality of the residuals was assessed using the Shapiro–Wilk test. The normality assumption was violated in several tests; however, for sample sizes where the number of observations per variable is >10, violations of this normality assumption often do not noticeably affect the results, and correcting for this normality violation in the regression analysis is not recommended [86]. The condition of more than 10 observations for each variable was met in our sample.

For multiple comparisons, FDR-adjusted *p*-values < 0.05 were considered statistically significant. The exception was the MLR analysis. Because the study was exploratory and the MLR analysis included multiple statistical tests (*n* = 105 for R-squared and *n* = 399 for variables) that were not completely independent (Figure 6A,C), we set FDR < 0.1 for R-squared and covariate *p*-values to avoid missing discoveries in this part of the study [87,88].

## 5. Conclusions

Our exploratory exome-based study is the first to consider and compare RBC-related phenotype–genotype correlations in chronic and acute critical illnesses (msTBI and COVID-19). Analysis of the WES data, focusing on rare HI variants in specific gene sets, yielded both similar and different results in the two patient cohorts, supporting the view that genetic factors contribute to RBC-related phenotypic performance in severe and heterogeneous health conditions. The most interesting result was the association of Notch signaling pathway genes with the development of anemia in both patient cohorts; however, further research is needed.

The long path from genotype to phenotype is influenced by many factors that contribute to the severity of the phenotypic manifestation, and dissecting these factors can help guide the selection of a treatment strategy. For example, in inflammation, genetically mediated downregulation of the Notch signaling gene *LFNG* can lead to blood–brain barrier dysfunction and has been associated with the cerebrospinal fluid (CSF) volume [89]. Recent work has highlighted the importance of blood, RBC, and CSF dynamic parameters, such as compliance in blood–brain–CSF interactions in patients with TBI [90]. Future studies should integrate genetic data with other physiological variables, such as CSF, blood, and RBC dynamic parameters, to provide a more comprehensive approach to TBI research by elucidating the role of genetics in the transition from an endophenotype to a clinical phenotype.

## Figures and Tables

**Figure 1 ijms-26-01239-f001:**
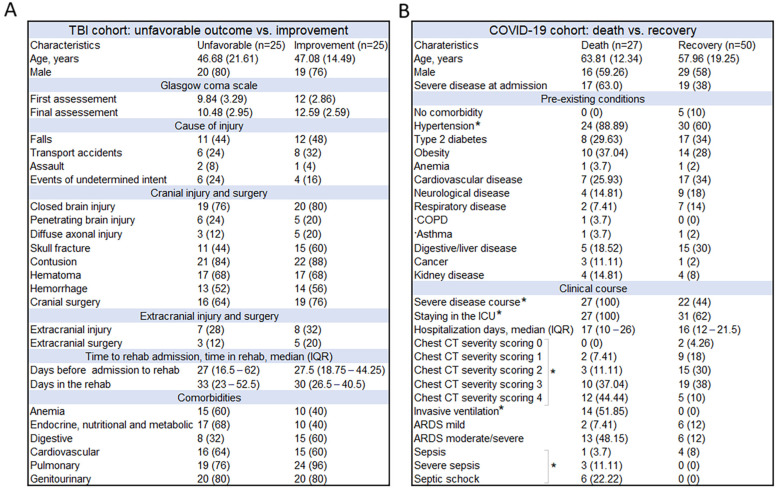
The characteristics of the patient cohorts: (**A**) TBI cohort and (**B**) COVID-19 cohort. Significant results after FDR correction are marked with an asterisk.

**Figure 2 ijms-26-01239-f002:**
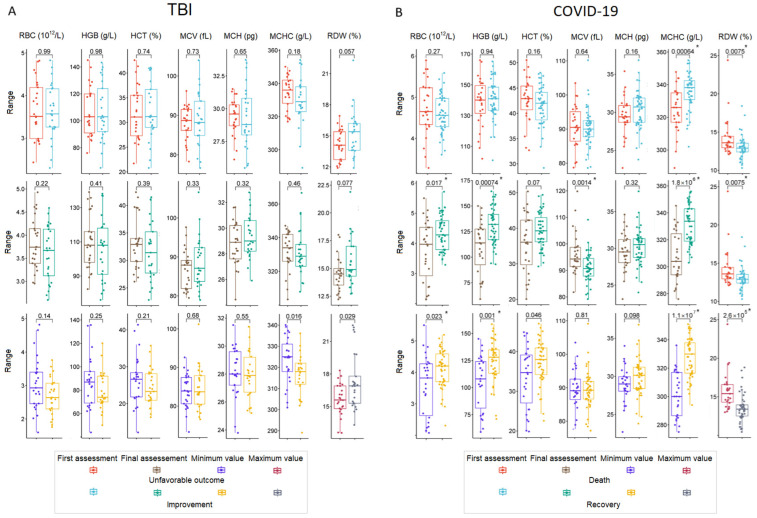
Box plots with *p*-value comparing RBC-related parameters at first, final, and min–max measurements in patients with different outcomes in the TBI cohort (**A**) and COVID-19 cohort (**B**). Significant results after FDR correction are marked with an asterisk.

**Figure 3 ijms-26-01239-f003:**
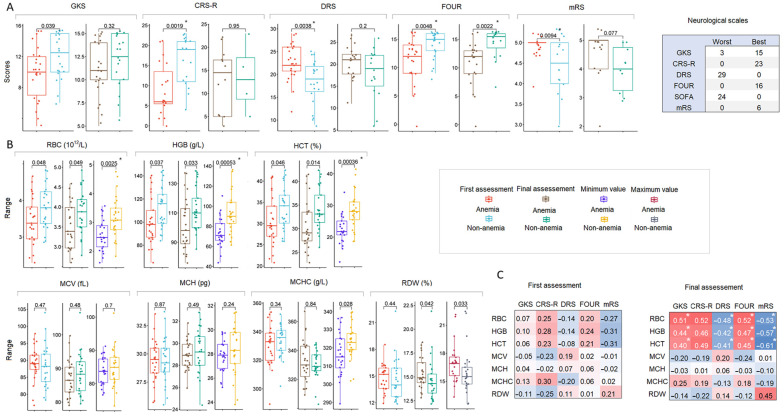
TBI patients with and without anemia: RBC-related parameters and neurological scale scores. (**A**) Box plots with *p*-values comparing neurological scale scores at the first and final assessments. (**B**) Box plots with *p*-values comparing RBC-related parameters at the first, final, and min-max measurements. (**C**) Heat map matrix of Spearman’s correlation coefficients for neurological scale scores and RBC-related parameters at the first and final assessments. Significant results after FDR correction are marked with an asterisk. Abbreviations: Glasgow Coma Scale (GCS), Coma Recovery Scale-Revised (CRS-R), Disability Rating Scale (DRS), Full Outline of UnResponsiveness (FOUR) scale, and modified Rankin Scale (mRS).

**Figure 4 ijms-26-01239-f004:**
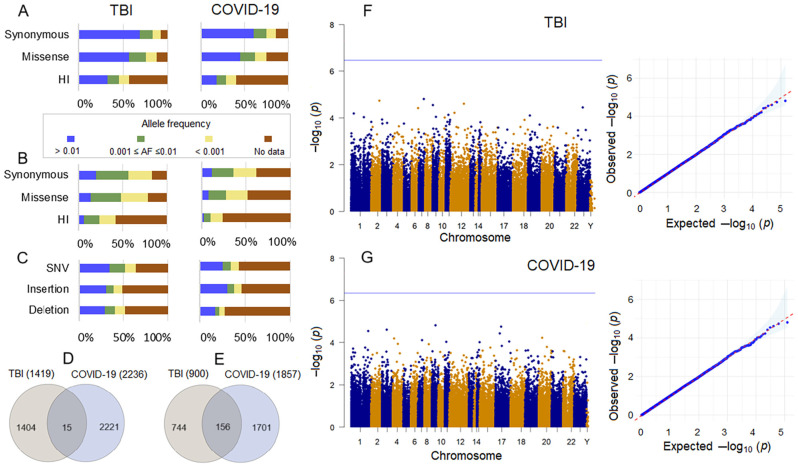
Exome data. (**A**) Distribution of HI, missense, and synonymous variants by allele frequency (AF), according to GnomAD. (**B**) Distribution of singletons by AF. (**C**) Distribution of variant types by AF. (**D**) Venn diagram of the QVs in the TBI and COVID-19 variant sets. (**E**) Venn diagram of genes with QVs in the TBI and COVID-19 gene sets. (**F**) Manhattan (left) and QQ (right) plots of the association *p*-value for anemia in the TBI sample. The Bonferroni-corrected significant *p*-value of 3.47 × 10^−7^ is indicated by the blue line. (**G**) Manhattan (left) and QQ (right) plots of the association *p*-value for HGB < 100 g/L at the final measurement in the COVID-19 sample. The Bonferroni-corrected significant *p*-value of 4.75 × 10^−7^ is marked by the blue line.

**Figure 5 ijms-26-01239-f005:**
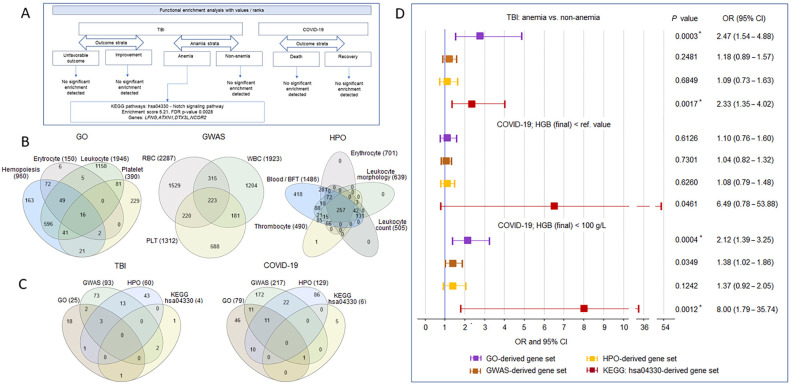
Generating and analyzing gene sets. (**A**) The results of “Proteins with Values/Ranks–Functional Enrichment Analysis” for the gene sets with QVs in the TBI and COVID-19 strata. (**B**) Venn diagrams for GO-, GWAS-, and HPO-derived gene sets for hematology-related terms, blood cell phenotypes, and abnormalities of blood and blood-forming tissues (Blood/BFT) and blood cells, respectively. (**C**) Venn diagrams representing GO-, GWAS-, HPO-, and KEGG hsa04330-derived gene sets with QVs in the TBI and COVID-19 samples. (**D**) Association analysis of the cumulative effects of QVs in the gene sets of interest in the TBI and COVID-19 samples. Unadjusted two-tailed Cochran–Mantel–Haenszel exact *p*-values, odds ratios, and horizontal bars indicating 95% confidence intervals are shown. Significant results after FDR correction are marked with an asterisk.

**Figure 6 ijms-26-01239-f006:**
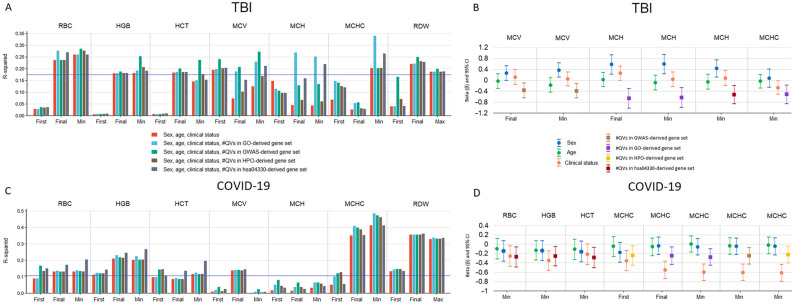
Genetic effects on RBC-related parameters in multiple linear regression (MLR) analysis. (**A**,**C**) Percentage of variance explained in RBC-related parameters for the MLR model of non-genetic and genetic predictors in the TBI (**A**) and COVID-19 (**C**) samples. In the TBI sample, clinical status at the first measurement was defined as the GCS score at the first assessment, and at the final and minimum measurements as unfavorable outcome or improvement. In the COVID-19 sample, clinical status at the first measurement was defined as COVID-19 severity at admission, and death or recovery at the final and minimum measurements. The FDR-corrected (threshold 0.1) significant R-squared level is indicated by a blue line. (**B**,**D**) FDR-corrected (threshold 0.1) significant standardized regression coefficients (beta) for genetic variables represented by the number of QVs per person in the studied gene sets in combination with non-genetic variables for RBC-related parameters in the TBI (**B**) and COVID-19 (**D**) samples. Vertical bars indicate 95% confidence intervals for beta. Dummy variables: sex—male 1 and female 2; TBI sample—improvement 1 and unfavorable outcome 2; COVID-19 sample—severity of COVID-19 at admission: non-severe 0 and severe 1; and COVID-19 outcome—recovery 0 and death 1.

## Data Availability

Data for TBI patients are only available upon request from the corresponding author due to privacy or ethical restrictions. All raw sequencing data for COVID-19 patients have been submitted to the NCBI BioProject database (https://www.ncbi.nlm.nih.gov/bioproject/ (accessed on 10 October 2024)) under accession number PRJNA947511.

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
