# Peer review of "Red Blood Cell-Related Phenotype–Genotype Correlations in Chronic and Acute Critical Illnesses (Traumatic Brain Injury Cohort and COVID-19 Cohort)"

_ijms, 2025, doi:10.3390/ijms26031239_

Round 1

Reviewer 1 Report

Comments and Suggestions for Authors

The manuscript "Red blood cell-related phenotype-genotype correlations in chronic and acute critical illness (TBI cohort and COVID-19 cohort)“ explores the role of rare functional variants in specific gene sets in anemia and RBC-related phenotypes in patients with severe chronic and acute conditions, represented by TBI and COVID-19 patients. The study provides valuable insights into the interplay between genetic factors, RBC function, and disease outcomes. After going through the manuscript, I have following comments for the authors:

1.     Were the known variants in RBC-related genes (such as ANK1, SPTA1) and genetic markers linked to RBC and hemoglobin function (e.g.HBB, HBA1) investigated?

2.     Please briefly discuss the personalized treatment strategies (eg. Transfuion strategies, targeted therapies) based on the results of the study.

Comments on the Quality of English Language

Language is fine. Minor grammatical corrections and syntax adjustments suggested.

Author Response

Dear Reviewer,

Please see our answers to your questions in the attached file.

Comments on the Quality of English Language

Language is fine. Minor grammatical corrections and syntax adjustments suggested.

Response to comments on the Quality of English Language

Dear Reviewer, thank you for pointing this out. The manuscript has been carefully read and changes have been made.

Best regards, Lyubov Salnikova

Reviewer 2 Report

Comments and Suggestions for Authors

I sincerely appreciate each new information about the management of this life-threatening clinical condition, such as COVID-19.

Content suggestions:

1.         How will these results influence the treatment of the patients ?

2.         Can the Authors provide the information about the markers of iron metabolism, e.g. ferririn level associated with anaemia and also studied illnesses ?

Author Response

Dear Reviewer,

Please see our answers to your questions in the attached file.

Best regards, Lyubov Salnikova
